

# CoMiniGut—a small volume *in vitro* colon model for the screening of gut microbial fermentation processes

Maria Wiese[1], Bekzod Khakimov[1,2], Sebastian Nielsen[3], Helena Sørensen[1], Frans van den Berg[1] and Dennis Sandris Nielsen[1]

[1] Department of Food Science, University of Copenhagen, Frederiksberg, Denmark
[2] Department of Plant and Environmental Sciences, Copenhagen Plant Science Center, University of Copenhagen, Frederiksberg, Denmark
[3] Plant Facilities and Workshops, University of Copenhagen, Frederiksberg, Denmark

Corresponding author
Maria Wiese, mwiese@food.ku.dk

## ABSTRACT

Driven by the growing recognition of the influence of the gut microbiota (GM) on human health and disease, there is a rapidly increasing interest in understanding how dietary components, pharmaceuticals and pre- and probiotics influence GM. *In vitro* colon models represent an attractive tool for this purpose. With the dual objective of facilitating the investigation of rare and expensive compounds, as well as an increased throughput, we have developed a prototype *in vitro* parallel gut microbial fermentation screening tool with a working volume of only 5 ml consisting of five parallel reactor units that can be expanded with multiples of five to increase throughput. This allows e.g., the investigation of interpersonal variations in gut microbial dynamics and the acquisition of larger data sets with enhanced statistical inference. The functionality of the *in vitro* colon model, Copenhagen MiniGut (CoMiniGut) was first demonstrated in experiments with two common prebiotics using the oligosaccharide inulin and the disaccharide lactulose at 1% (w/v). We then investigated fermentation of the scarce and expensive human milk oligosaccharides (HMOs) 3-Fucosyllactose, 3-Sialyllactose, 6-Sialyllactose and the more common Fructooligosaccharide in fermentations with infant gut microbial communities. Investigations of microbial community composition dynamics in the CoMiniGut reactors by MiSeq-based 16S rRNA gene amplicon high throughput sequencing showed excellent experimental reproducibility and allowed us to extract significant differences in gut microbial composition after 24 h of fermentation for all investigated substrates and fecal donors. Furthermore, short chain fatty acids (SCFAs) were quantified for all treatments and donors. Fermentations with inulin and lactulose showed that inulin leads to a microbiota dominated by obligate anaerobes, with high relative abundance of Bacteroidetes, while the more easily fermented lactulose leads to higher relative abundance of Proteobacteria. The subsequent study on the influence of HMOs on two infant GM communities, revealed the strongest bifidogenic effect for 3′SL for both infants. Inter-individual differences of infant GM, especially with regards to the occurrence of Bacteroidetes and differences in bifidobacterial species composition, correlated with varying degrees of HMO utilization foremost of 6′SL and 3′FL, indicating species and strain related differences in HMO utilization which was also reflected in SCFAs concentrations, with 3′SL and 6′SL resulting in significantly higher butyrate production compared to 3′FL. In conclusion, the increased throughput of CoMiniGut strengthens experimental conclusions through elimination of statistical

interferences originating from low number of repetitions. Its small working volume moreover allows the investigation of rare and expensive bioactives.

## INTRODUCTION

Growing evidence shows that human health and disease are linked to gut microbiota (GM) composition and functionality (*Marchesi et al., 2016*). This has led to an increasing interest in investigations on the effect of dietary components, pharmaceuticals, and pre- and probiotics on GM composition and function (*Payne et al., 2012*). Human intervention studies are the "gold standard" in the study of GM and GM-host interactions, but these are also expensive and hampered by ethical constrains. Similarly, animal models are essential for the study of e.g., the impact of GM on many diseases, but animal GM composition differs from the human GM and for ethical reasons the use of animal experiments should be minimised if alternatives are available (*Fenwick, Griffin & Gauthier, 2009*; *Krych et al., 2013*).

*In vitro* colon models facilitate investigations circumventing ethical constrains and lowering study costs. These models provide unlimited screening possibilities of new products, facilitating continuous monitoring and sampling possibilities under standardized conditions. A variety of *in vitro* colon models exists as reviewed previously (*Macfarlane & Macfarlane, 2007*; *Payne et al., 2012*; *Venema & Van den Abbeele, 2013*). The models range from batch to continuous cultures, consisting of single or multistage setups. The latter design facilitates the simulation of spatial, temporal, nutritional and physicochemical properties of the colon through a series of three connected chemostats, simulating the proximal, transverse and distal colon regions (*Payne et al., 2012*).

Along the colon a continuous increment in pH and lowered redox potential is seen due to a varying degree of SCFAs production, cross-feeding of SCFAs by GM members, absorption by the host and increased proteolysis (*Fallingborg et al., 1993*).

*In vitro* simulations of the colon which are operated with larger volumes do generally not facilitate investigations on the GM modulating effects of expensive compounds or novel synthesized materials only available in small amounts e.g., due to high costs. Furthermore, the usually low throughput-number of larger *in vitro* models with only a few or no technical replicates is reducing the statistical strength of experimental data and scientific conclusions.

HMOs are a family of structurally diverse unconjugated glycans, which constitute the third most abundant human milk component (*Newburg, 1996*). HMOs display resistance to the low pH in the infant's stomach as well as hydrolysis by host enzymes and gastrointestinal absorption (*Engfer et al., 2000*) and are selectively fermented by the infant GM. Variations in infant GM composition due to feeding and delivery mode have been described, and inter-individual differences in GM prevail even when grouped according to delivery and feeding mode (*Azad et al., 2013*). Characteristic for breast-fed infants is the selective nourishment

and support of a protective, co-evolved gut microbiota dominated by bifidobacteria and Bacteroides (*Bäckhed et al., 2005*; *Haarman & Knol, 2005*).

HMOs are composed of lactose reducing ends elongated with up to 25 *N*-acetyllactosamine units, which can be extensively fucosylated and/or sialylated through R-glycosidic linkages (*Ninonuevo et al., 2006*). This translates to over 200 different known human milk oligosaccharide (HMO) structures (*Ninonuevo et al., 2006*). Emerging data indicates that HMO profiles are dynamic, revealing fluctuations between and within lactation in individual mothers (*Ninonuevo et al., 2006*; *Thurl et al., 2010*; *Totten et al., 2012*). Knowledge on the relationship between the levels of specific milk oligosaccharides and their role in maternal and infant health and development is still scarce (*Zivkovic et al., 2011*). However, in general HMOs are associated with an array of health benefits, of which many are thought to be directly or indirectly associated with the infants gut microbiota (*Bode, 2012*). Further, HMO shaped microbial communities can aid to prevent the colonization by pathogens through competitive exclusion (*Morrow et al., 2004*). HMOs have also been reported to act as decoy pathogen binding sites directly inhibiting the adhesion and invasion of pathogenic microbes (*Marcobal et al., 2010*) modulating intestinal epithelial cell growth and infant immune system as reviewed previously (*Bode, 2006*; *Bode, 2012*; *German et al., 2008*).

The utilization of specific HMOs by the infant GM is still being explored. HMO consumption is conserved in some infant-associated types of bifidobacteria, such as the *B. longum* subsp. *infantis* lineage, whereas other display strain-specific phenotypic variation (*LoCascio et al., 2007*; *Ward et al., 2006*). Several studies have investigated the utilization of HMOs in anaerobic cultures with single species (*Marcobal et al., 2010*; *Sela et al., 2012*; *Sela et al., 2011*; *Ward et al., 2007*; *Ward et al., 2006*; *Yu, Chen & Newburg, 2013*), where e.g., *Marcobal et al. (2010)* reported the metabolization of HMOs by *Bacteroides fragilis* and *Bacteroides vulgatus* strains with high efficiency.

Functional HMO alternatives such as galacto-oligosaccharides (GOS) are produced by glycoside hydrolases (GH) using lactose as substrate (*Torres et al., 2010*) and fructo-oligosaccharides (FOS), can be produced through transfructosylation of sucrose or hydrolysis of inulin by endoinulinases (*Singh & Singh, 2010*), are the main prebiotics currently added to infant formula (*Rinne et al., 2005*).

With the assumption that the human colon can be simulated *in vitro* in small volumes, and with the dual objective of facilitating the investigation of rare and expensive compounds, as well as an increase of throughput, we have developed a prototype *in vitro* colon model with a working volume of only 5 ml mimicking an average colon transit time of 24 h. Multiples of 5-plex CoMiniGuts can be aligned and constitute, in combination with high throughput molecular biology techniques and advanced analytics, an efficient screening platform to investigate GM dynamics and fecal metabolomics. Here we demonstrate as proof of principle the use of CoMiniGut for the investigations of the two common, well studied prebiotics inulin and lactulose. We then investigate the fermentation of the rare and expensive compounds, and most original prebiotics, namely human milk oligosaccharides and compare the effect with that of fructo-oligosaccharides (*Bode, 2012*; *Marcobal et al., 2010*).

## MATERIAL AND METHODS

### The CoMiniGut

The CoMiniGut prototype consists of a climate box with five parallel single-vessel, stirred, anaerobic reactor units, which are pH monitored and controlled (Fig. 1). Each anaerobic reactor unit consists of a fused quartz glass vial (FQ-2010, Fused quartz crucible, cylindrical, 10 ml, OD 22 mm × H 33 mm, 5 ml working volume; AdValue Technology, Tucson, AZ, USA) positioned in a 150 ml polymethylmethacrylat (PMMA) compartment. Anaerobic conditions are achieved using either Anaerogen compact sachets (AN0020D; ThermoScientific, Waltham, MA, USA) positioned inside the PMMA compartments or via integrated gas in- and outlets for flushing the compartments with nitrogen (99.8%) to maintain anaerobiosis (this also facilitates gas or headspace sampling). Resazurin soaked indicators are used to signal anaerobiosis (Anaerobe Indicator Test; Sigma-Aldrich, St. Louis, MO, USA). The lid of the PMMA compartments is composed of a PMMA ring and an exchangeable vacuum greased silicon rubber septa (20420-U SUPELCO GR-2 Rubber Sheet Stockmaterial; Sigma-Aldrich, St. Louis, MO, USA) containing a pH probe inlet. The rubber septa are penetrable by needles for pH control, sampling, as well as feeding of substrate if desired. The parallel alignment of the five reactor vessels in one unit is based on a magnetic stirrer with five stirring positions. The climate box is kept at a temperature of 37 °C by a circulating water bath connected to a heat-exchange plate inside the climate box and an external temperature probe positioned in the box for feedback control. A ventilation system secures even temperature distribution, and temperature logging (Temp 101A MadgeTech Temperature data logger) is performed throughout experiments. The pH is monitored via a 6 channel pH meter and data logger (Consort multi-parameter analyser C3040). The pH meter is connected to a laptop running in-house Matlab scripts for pH control (ver. R2015a; The MathWorks, Inc., Natick, MA, USA), which regulates a multichannel syringe pump charged with syringes containing 1 M NaOH. The syringes (10 ml; BD Biosciences, San Jose, CA, USA) are connected via tubing (VWR) and injection needles (Frisenette, Knebel, Østjylland, Denmark) into the fermentation compartments.

### Fecal inoculum

For inulin and lactulose fermentations fecal samples from two healthy adults (F1, F2) were individually homogenized in a 1:1 ratio with 1M PBS/20% glycerol (v/v) in a stomacher bag for 2 × 60 s using the Stomacher (Stomacher 400; Seward, Worthing, UK) at normal speed. For HMO fermentations fecal samples were prepared as stated above for two infant fecal donors. Participants and parents of infants provided consent, sampling and use of fecal material for inoculation of the CoMiniGut fermentations has been approved by the Ethical Committee (E) for the Capital Region of Denmark (H-15001754).

Both infant donors were healthy males, approximately six months of age, born vaginally, have been exclusively breast-fed and did not receive antibiotic treatment nor probiotics. Fecal slurries, with a final glycerol concentration of 10% (a standard glycerol concentration used for fecal transplants (*Hamilton et al., 2012*)) were then aliquoted into cryo vials and stored at −60 °C until use.
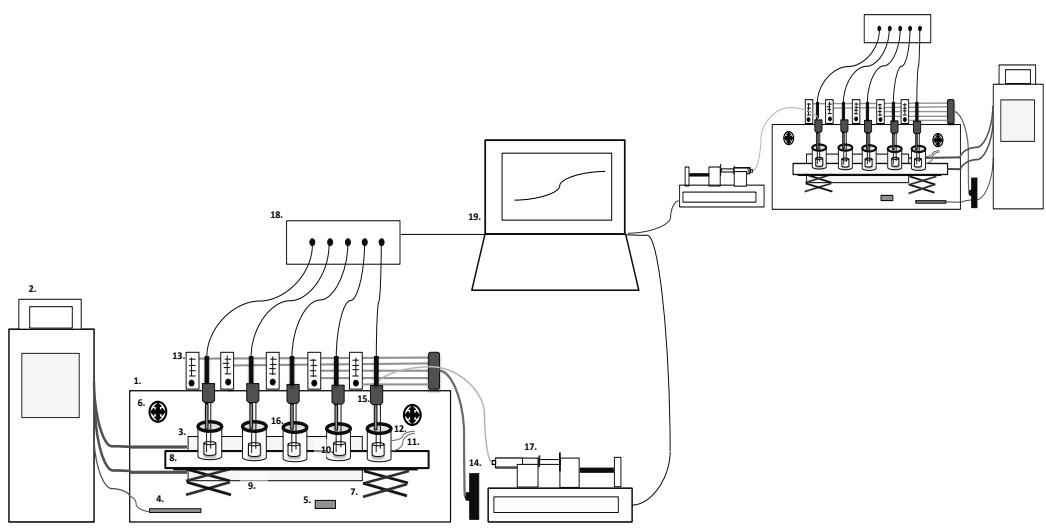

**Figure 1** **CoMiniGut model overview.** 1. Climate Box, 2. Water bath, 3. Heating plate, 4. Temperature probe, 5. Temperature data logger, 6. Ventilators, 7. Lab-jack table, 8. Magnetic stirrer, 9. PMMA compartment, 10. Reaction vial, 11. Gas inlet, 12. Gas outlet, 13. Flow meters, 14. Nitrogen/gas tap, 15. pH probe and needle inlets, 16. Lid with septa, 17. Multichannel syringe pump, 18. pH meter, 19. Computer.

Fecal glycerol stocks were thawed and further diluted with 0.1 M PBS pH 5.6, ratio 1:4, on the day of the experiment. CoMiniGut reaction vessels containing 4.5 ml of media were then inoculated with 0.5 ml of fecal slurry to achieve an inoculation at 10% of the fermentation volume (hence 1% original fecal matter), diluting the glycerol out to 0.2% (v/v). The viability of feces derived microbiota of frozen and fresh fecal slurries was evaluated through the counting of colony forming units (CFUs) on GAM and BHI agar plates after anaerobic incubation for 72 h at 37 °C.

### *In vitro* fermentation media and conditions

Stirred batch-culture fermentations (5 ml working volume) were set up and aseptically filled with 4.5 ml basal medium (0.5 g/l bile salts, 2 g/l peptone water, 2 g/l yeast extract (all purchased from Oxoid), 0.1 g/l NaCl, 0.04 g/l $K_2HPO_4$, 0.04 g/l $KH_2PO_4$ (all purchased from Merck), 0.01 g/l $MgSO_4 \times 7H_2O$, 0.01 g/l $CaCl_2 \times 6H_2O$, 2 g/l $NaHCO_{3-}$, Hemin 0.002 g/l, Vitamin $K_1$ 10 μl, Tween-80 2 ml (all purchased from Sigma-Aldrich), 0.5 g/l L-Cysteine HCl (Calbiochem, San Diego, CA, USA)). Sigma colon medium was chosen for screening purposes as also done previously (*Fooks & Gibson, 2002*; *Sanz, Gibson & Rastall, 2005*; *Sarbini et al., 2013*; *Saulnier, Gibson & Kolida, 2008*).

For the fermentations of inulin and lactulose the media was supplemented with 50 mM MES 2-(*N*-morpholino) ethanesulfonic acid (Sigma-Aldrich, St. Louis, MO, USA) buffer, divided into two batches and one batch was supplemented with 1% inulin (w/v) from chicory (Sigma-Aldrich) and the other batch with 1% lactulose (w/v) (Sigma-Aldrich). For the HMO fermentations basal colon media supplemented with 50 mM MES (Sigma-Aldrich) buffer, was divided into five batches and four batches were supplemented with each one of the four HMOs 1% (w/v). HMO used were 3-Fucosyllactose (3FL)
(OligoTech, Crolles, France), 3-Sialyllactose (3SL) and 6-Sialyllactose (6SL) (Carbosynth, Compton, UK), Oligofructose (FOS) (Orafti P95; Oreye, Belgium) and a carbohydrate negative control. The fermentations were performed for each substrate and fecal sample in quadruplicates. Fermentations were performed with pH control. The pH was set to increase from 5.7 to 6.0 during the first 8 h of fermentation simulating pH conditions prevalent in the proximal colon, followed by an 8 h pH increment from pH 6.0 to 6.5 representing the pH conditions in the transverse colon and finally a pH increment from 6.5 to 6.9 for distal colon. Samples were taken for endpoint analysis after 24 h of fermentation as previously described (*Glei et al., 2006*; *Kneifel, 2000*; *Stein et al., 2011*). The HMOs were fermented in quadruplicates, the negative control without HMOs are represented in triplicates. GraphPad Prism software (Version 6.0) was used for statistical analyses of the pH controlled fermentations.

## DNA extraction
One ml of each fermentation endpoint (at 24 h) was pelleted via centrifugation at 13.000 g for 10 min and gDNA was extracted from the pellet using the Power Soil Kit protocol (MoBio Laboratories, Carlsbad, CA, USA). The FastPrep bead-beating step was performed in 3 cycles of 15 s each at a speed of 6.5 M/s in a FastPrep-24$^{TM}$ Homogenizer (MP). DNA quantity and quality were measured using a NanoDrop 1000 (Thermo Scientific, Waltham, MA, USA).

## 16S rRNA gene library preparation
The fecal microbiota composition of *in vitro* fermentation samples were determined using tag-encoded 16S rRNA gene MiSeq-based (Illumina, CA, USA) high throughput sequencing. The V3 region of the 16S rRNA gene was amplified using primers compatible with the Nextera Index Kit (Illumina, San Diego, CA, USA) NXt_338_F: 5′-TCGTCGGCAGCGTCAGATGTGTATAAGAGACAGACWCCTACGGGWGGC AGCAG-3′ and NXt_518_R: 5′-GTCTCGTGGGCTCGGAGATGTGTATAAGAGACAGA TTACCGCGGCTGCTGG-3′ (*Ovreås et al., 1997*) the PCR reactions and library preparation was conducted as described in *Kristensen et al. (2016)*.

## High throughput sequencing and data treatment
The raw dataset containing pair-ended reads with corresponding quality scores were merged and trimmed using *fastq_mergepairs* and *fastq_filter* scripts implemented in the UPARSE pipeline. The minimum overlap length was set to 10 base pairs (bp). The minimum length of merged reads was 150 bp, the maximum expected error $E$ was 2.0, and the first truncating position with quality score was $N \leq 4$. Purging the dataset from chimeric reads and constructing *de novo* Operational Taxonomic Units (OTU) were conducted using the UPARSE pipeline (*Edgar, 2013*). The Green Genes (13.8) 16S rRNA gene collection was used as a reference database (*McDonald et al., 2012*). Quantitative Insight Into Microbial Ecology (QIIME) open source software (*Caporaso et al., 2010*) (1.7.0 and 1.8.0) was used for the subsequent analysis steps. Principal coordinate analysis (PCoA) plots were generated with the Jackknifed Beta Diversity workflow based on 10 UniFrac distance metrics calculated using 10 subsampled OTU tables. The number of sequences taken for each jackknife subset

was set to 90% of the sequence number within the most indigent sample, hence 77,000 reads per sample for the inulin and lactulose 16S rRNA gene amplicon sequencing based analysis and 87,000 reads/sample for the HMO 16S rRNA gene amplicon sequencing based analysis. Analysis of similarities (ANOSIM) was used to evaluate group differences using weighted and unweighted (*Lozupone & Knight, 2005*) UniFrac distance metrics that were generated based on rarefied (77,000 reads/sample and 87,000 reads/sample, respectively) OTU tables. The relative distribution of the genera registered was calculated for unified and summarized in genus level OTU tables. Alpha diversity measures expressed as observed species values (sequence similarity 97%) were computed for rarefied OTU tables (77,000 reads/sample and 87,000 reads/sample respectively) using the alpha rarefaction workflow. Differences in alpha diversity were determined using a *t*-test-based approach employing the non-parametric (Monte Carlo) method (999 permutations) implemented in the compare alpha diversity workflow. The ANOVA determined significance of quantitative (relative abundance) association of OTUs with given categories, $p$ values were False Discovery Rate (FDR) corrected. These were calculated based on 1,000 subsampled OTU-tables rarefied to an equal number of reads (77,000 reads/sample and 87,000 reads/sample, respectively).

## SCFA extraction and analysis

SCFAs profiles were extracted from 1 ml of pH controlled fermentations. In brief: 2 ml of 0.3 M oxalic acid was added to the sample and vortexed for 1 min, followed by centrifugation at 2,800 g for 15 min. Subsequently 800 µl was filtered through 0.45 µm pore size Ultrafree-MC-HV filters (Millipore, Cork, Ireland). 600 µl of the filtrate was transferred into HPLC vials containing 30 µl of the internal standard 50 mM 2 ethyl butyrate (Sigma-Aldrich), in water. The samples were then analyzed using GC-MS (Agilent 7890A GC and an Agilent 5973 series MSD; Agilent, Waldbronn, Germany). GC separation was performed on a Phenomenex Zebron ZB-WAXplus column (30 m × 250 µm × 0.25 µm). A sample volume of 1 µl was injected into a split/split-less inlet at 285 °C using a 2:1 split ratio. Septum purge flow and split flow were set to 13 ml/min and 2 ml/min, respectively. Hydrogen was used as carrier gas, at a constant flow rate of 1.0 ml/min. The GC oven program was as follows: initial temperature 100 °C, equilibration time 1.0 min, heat up to 120 °C at the rate of 10 °C/min, hold for 5 min, then heat at the rate of 40 °C/min until 230 °C and hold for 2 min. Mass spectra were recorded in Selected Ion Monitoring (SIM) mode and the following m/z ion were detected at a dwell time of 50 ms: 41, 43, 45, 57, 60, 73, 74, 84 and the MS detector was switched off during the 1 min of solvent delay time. The transfer line, ion source and quadrupole MS temperatures were set to 230, 230 and 150 °C, respectively. The mass spectrometer was tuned according to manufacturer's recommendation using perfluorotributylamine (PFTBA). Dilution series of SCFAs standards of acetic, propionic, butyric, isobutyric, 2-methyl isobutyric, valeric and isovaleric acid (Sigma-Aldrich) were prepared at the concentrations of 1.000, 0.500, 0.250, 0.125, 0.060 and 0.030 mM for the construction of standard curves for quantification. For the analysis of SCFAs in the HMO fermentations, the amounts were subtracted by the SCFAs produced in the control fermentations, in order to extract the HMO effect.

## SCFAs data analysis

Initial analysis and visualization of the GC-MS data was performed using MSD ChemStation software (version E.02.02.1431; Agilent Technologies, Inc., Waldbronn, Germany). Mass spectra of SCFAs were compared against the NIST11 library (NIST, Gaithersburg, MD, USA). SCFAs peak areas were integrated from SIM chromatograms using in-house scripts written in Matlab (ver. R2015a; The MathWorks, Inc., Natick, MA, USA). Two SCFAs, 2-methyl isobutyric acid and isovaleric acid, co-eluted at the retention time range of 4.22–4.45 min; peak areas were determined by de-convoluting these peaks using base peaks at m/z ion 74 for 2-methyl isobutyric acid and m/z ion 60 for isovaleric acid. $T$-test and one-way ANOVA with post-hoc tests (significance level 0.05) were performed using GraphPad Prism.

# RESULTS

## Fermentations of inulin and lactulose

The pH-profiles of the inulin and lactulose fermentations with and without pH control are displayed in Fig. 2. The pH controlled fermentations were conducted with an average standard deviation of 0.07–0.10 pH units between experimental replicates over the course of 24 h, with no significant difference between F1 and F2 fermentations for each treatment (ANOVA, $p = 0.4$). Uncontrolled fermentations displayed an average standard deviation ranging from 0.23 to 0.36 pH units and differed significantly (ANOVA $p < 0.001$). The uncontrolled pH profiles displayed inter-individual differences as well as differences between fermentation substrates.

16S rRNA gene amplicon sequencing analysis yielded 7.796.212 reads with an average of $194.905.3 \pm 80.366.8$ reads per sample fulfilling the quality control requirements (minimum sequence length $\geq$ 180 bp, minimum average quality score $\geq$ 25). The alpha diversity of the fecal donor GM used as fermentation inoculum in this study did not differ significantly between the samples, with the number of observed species for F1 being 393 and for F2 318. The mean phyla composition was dominated by Firmicutes (F1: 59.7%, F2: 46.7%) and Bacteroidetes (F1: 37.2%, F2: 51.1%) for both donors with Cyanobacteria, Tenericutes, Verrumicrobia, Proteobacteria, Actinobacteria, Lentisphaerae, and Fusobacteria all constituting minor parts of the inoculum (0.01–1.5%). Principal Coordinates Analysis (PCoA) revealed a clear separation between the GM of the fecal inoculum prior and after *in vitro* fermentation (Fig. 3). Experimental replicates clustered together and statistical analysis on unweighted and weighted UniFrac matrices revealed significant separation between the treatments (ANOSIM unweighted inulin and lactulose F1 and F2 $R$-stat 0.1948, $p$-value 0.039; ANOSIM weighted inulin and lactulose F1and F2 $R$-stat 0.931, $p$-value 0.001) and displayed a clear donor effect, (Fig. 3). The relative abundance of Bacteroidetes differed significantly with 65% of the reads constituting Bacteroidetes for inulin versus 12% for lactulose ($p < 0.001$). The abundance of Proteobacteria also differed significantly with a mean relative abundance of 30% for inulin vs 79% for lactulose fermentations ($p < 0.001$) (Fig. 4A). The bifidogenic effect of the two tested prebiotics differed between the two tested GMs, with bifidobacterial abundance being significantly higher in inulin fermentations

A

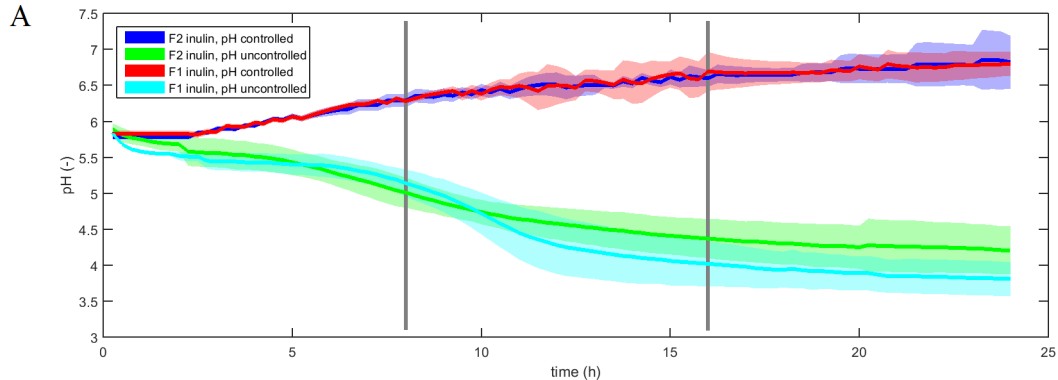

B

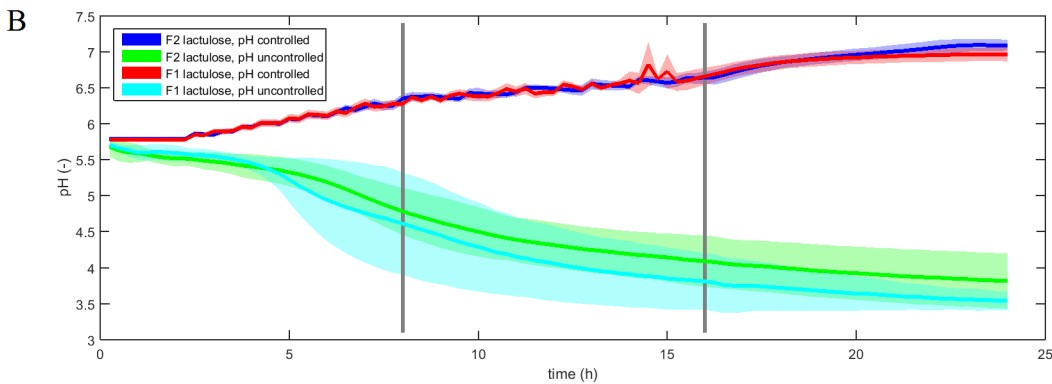

**Figure 2  pH profiles of inulin and lactulose fermentations.** Fermentation pH profiles of controlled and uncontrolled inulin (A) and lactulose (B) (1% w/v) fermentations using fecal inocula from adult donors F1 and F2 ($n = 4$, line is mean value, shadow area is convex hull).

for the GM of F2 (inulin 0.5% vs. lactulose 0.07%, $p = 0.005$), while for F1 the opposite picture was seen with a stronger (though non-significant) bifidogenic effect for lactulose (inulin 0.6% and lactulose 7.2%).

Seven SCFAs were successfully identified in the samples at a concentration range of 0.06–34 mM. In addition, one unknown peak at component retention time 5.85 min, was consistently detected in all samples (Fig. S1). It was not possible to identify this compound using the metabolite library in NIST11 (details found in Fig. S2). Individual GMs within the lactulose treatment, differed significantly in production of propionic ($p < 0.0001$) and acetic acid ($p < 0.0001$) (Fig. 4B). When fermenting inulin, the donor GM showed no significant difference in propionic and butyric acid production. The individual average SCFAs profiles (%) of the fermentations are shown in Fig. 4B.

SCFA production with inulin and lactulose as substrates differed for both donors with regard to butyric acid levels, which were higher in inulin fermentations ($1.28 \pm 0.59$ vs $0.06 \pm 0.05$ mM, $p = 0.003$). Similarly, more isobutyric acid ($0.12 \pm 0.00$ vs $0.09 \pm 0.01$ mM, $p = 0.05$) and propionic acid ($6.02 \pm 1.90$ vs $3.32 \pm 1.24$ mM, $p = 0.006$) was detected in inulin fermentates compared to the lactulose fermentates.

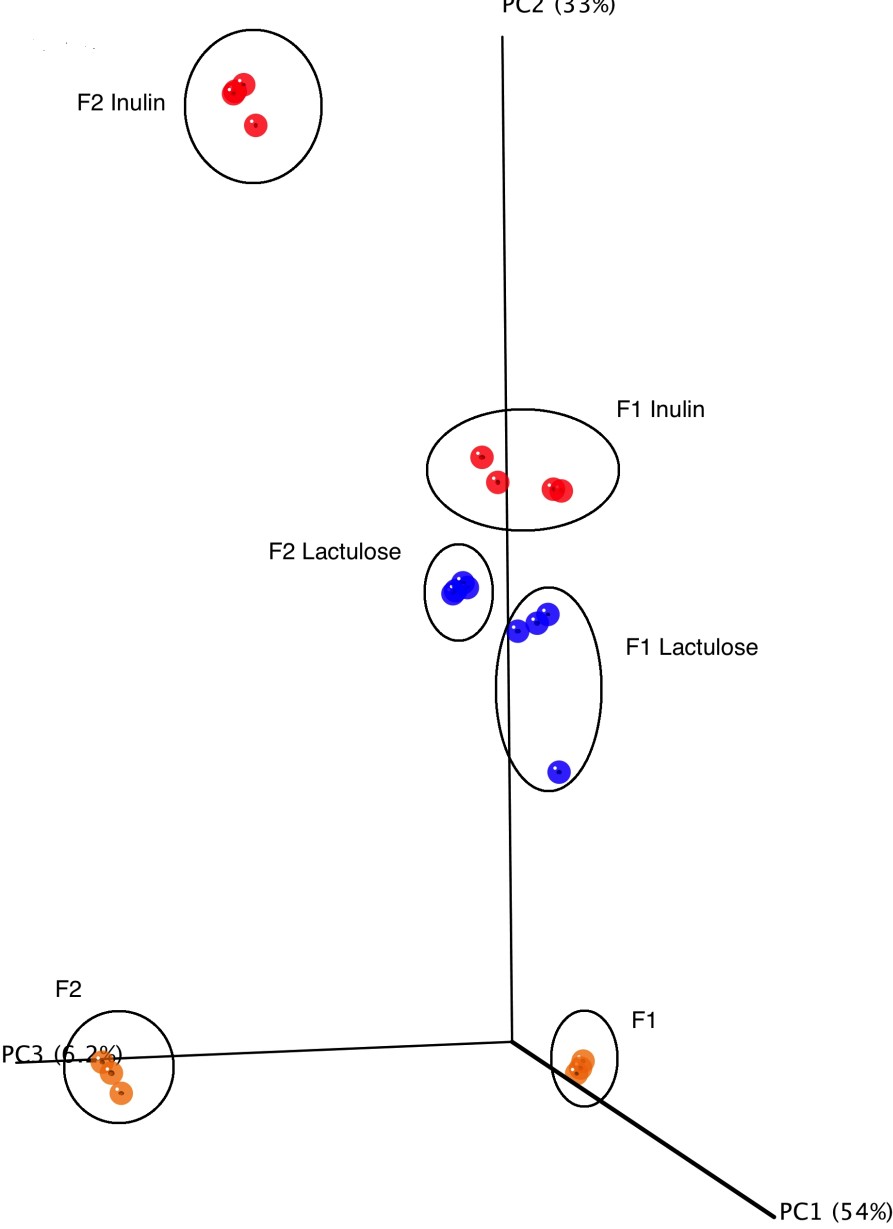

**Figure 3** **PCoA plot of 16S rRNA gene sequences.** PCoA score plot of 16S rRNA gene tag-encoded sequence reads based on weighted UniFrac distance metrics ($n = 4$, for all six experimental conditions).

## Fermentations of HMOs

Alpha diversity of the infant fecal samples did not differ significantly ($t$-stat $p = 0.08$). The GM composition of both infant donors at the phyla level was dominated by Actinobacteria $p < 0.01$ (B1 73.12%, B2 96.70%), but differed significantly in other phyla abundance levels. The GM of baby 2 feces had relatively low abundance of Firmicutes (B1 14.40%, B2 2.41%, $p < 0.01$) and Proteobacteria (B1 9.00%, B2 0.88%, $p < 0.01$) Bacteroidetes

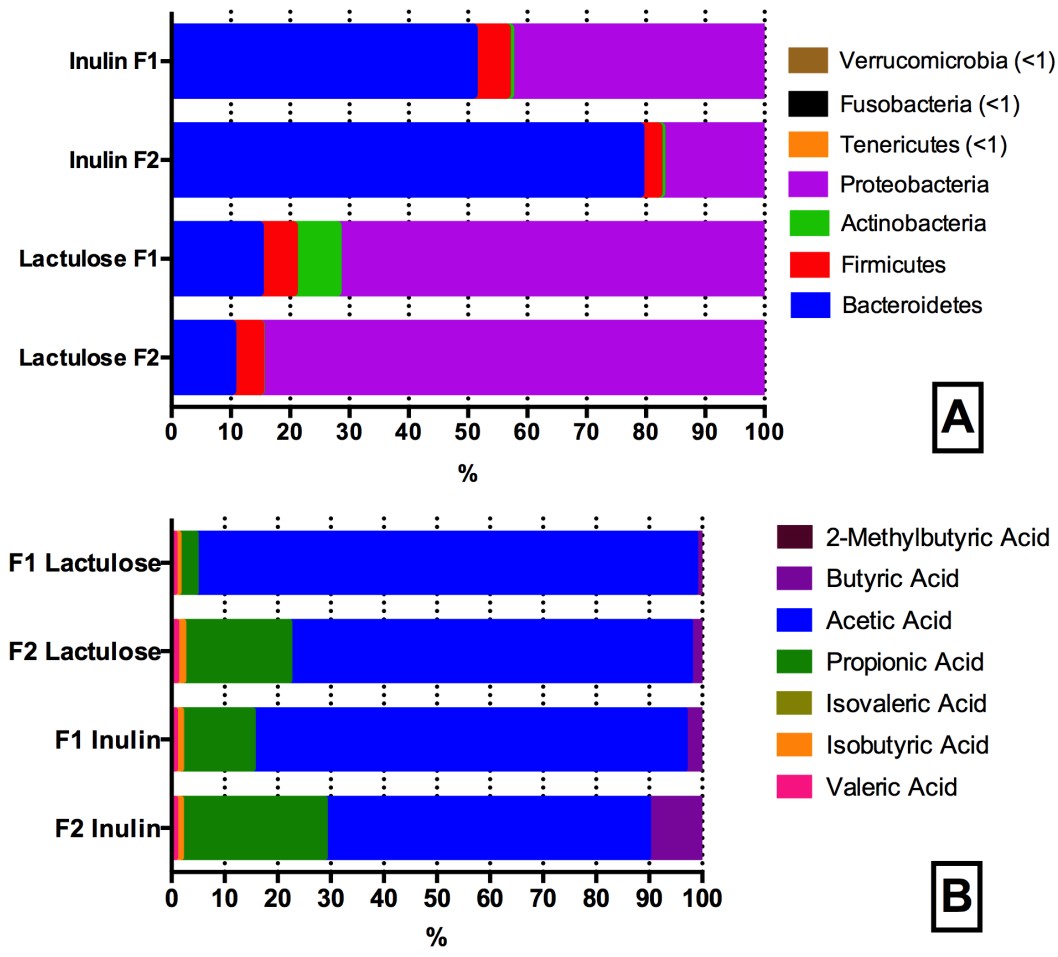

**Figure 4 Phyla (A) and SCFA (B) composition of the inulin and lactulose fermentations.** (A) Phyla composition (as determined by 16S rRNA gene amplicon sequencing) of pH controlled fermentations (24 h) of inulin and lactulose (1% w/v) using fecal inocula from adult donors F1 and F2 in CoMiniGut. ANOVA analysis of F1 and F2 inulin fermentations revealed significant differences (phylum level) in the relative abundances of Bacteroidetes, Proteobacteria and Firmicutes ($p < 0.01$), while no significant differences were detected between donors for Fusobacteria ($p = 0.1$) and Actinobacteria ($p = 0.5$). Variation between technical replicates was not significant ($n = 4$, F1 $p = 0.4$, F2 $p = 0.4$). For lactulose fermentations all phyla differed significantly between the two donors ($p \leq 0.05$), while variation between technical replicates was not significant for both donors, ($n = 4$, F1 $p = 0.6$; F2 $p = 0.8$). (B) Relative concentrations of short chain fatty acid profiles of pH controlled CoMiniGut fermentations with inulin and or lactulose (1% w/v) using fecal inocula from adults donors F1 and F2 ($n = 4$).

represented 0.009% of the relative abundance in baby 2 whereas they represented 3.25% of the reads in baby 1.

Of the investigated HMOs 3′SL clearly had the strongest bifidogenic effect and induced the highest relative abundance (21.5%) of Actinobacteria after 24 h of fermentation followed by 3′FL (9.9%) and 6′SL (8.4%), 2.4% in FOS and 5.7% in the control, the abundances differed significantly between treatments, $p = 0.03$ (Fig. 5).

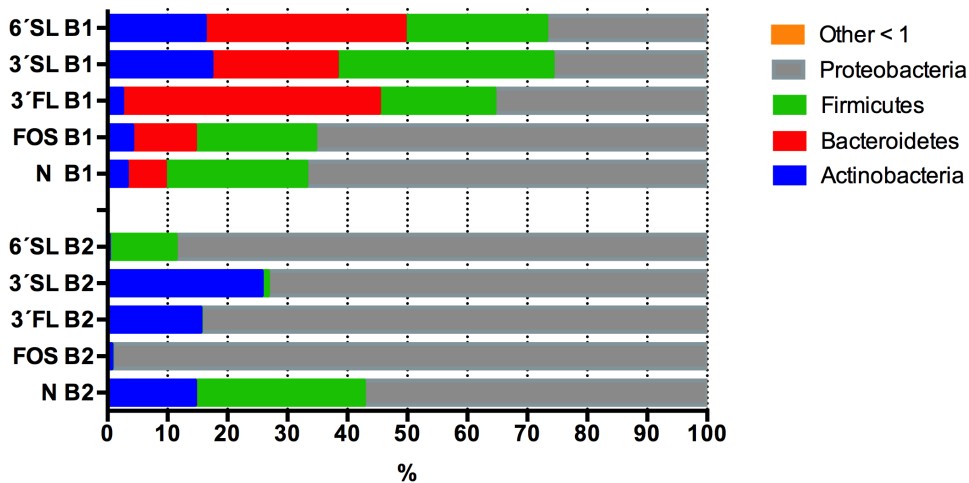

**Figure 5 Average Phyla composition of HMO fermentations.** Average Phyla composition, (relative % fraction of total) all fermentations for baby 1 and baby 2 with 3'SL, 6'FL, 6'SL and FOS as substrates as determined by 16S rRNA gene amplicon sequencing. 3'FL fermentations differed significantly in the relative abundance of assigned phyla between donors $p \leq 0.02$, but no significant differences were found between technical replicates ($n = 4$, B1 $p = 0.7$, B2 $p = 0.6$). For 3'SL fermentations significant differences between donors were detected for Proteobacteria and Firmicutes $p < 0.01$, whereas no significant variation was detected between technical replicates ($n = 4$, B1 $p = 0.8$, B2 $p = 0.6$). For 6'SL fermentations, relative abundance were significantly different for all phyla $p \leq 0.05$, whereas no significant variation was detected for technical replicates ($n = 4$, B1 $p = 0.8$, B2 $p = 0.5$). For FOS significant differences were detected for Bacteroidetes, Firmicutes and Proteobacteria $p \leq 0.05$, whereas no significant differences were detected between technical replicates ($n = 4$, B1 $p = 0.8$, B2 $p = 0.5$). For the negative control no significant differences between phyla composition were detected between donors $\geq 0.3$ and technical replicates ($n = 4$, B1 $p = 0.6$, B2 $p = 0.3$).

Looking at the babies separately, the highest abundances of Actinobacteria were identified in fermentations supplemented with sialyated HMOs for baby 1 with 3'SL 17.3% and 16.2% for 6'SL supplemented fermentations (Fig. 5). In the fermentations of HMOs inoculated with GM from baby 1 higher abundances of Actinobacteria were observed in all supplemented media compared to the negative control with the exception of 3'FL. The 3'FL fermentation also displayed the highest abundance of Bacteroidetes (42.8%), while Firmicutes were most abundant in the 3'SL fermentations (35.9%). Proteobacteria were lower in HMO supplemented treatments compared with FOS and non-supplemented control (Fig. 5). In fermentations inoculated with baby 2 GM, which barely contained representatives of the phyla Bacteroidetes, the three sialyated supplemented fermentations displayed the highest relative abundance of 25.7% Actinobacteria. Significant differences in Actinobacteria were detected between treatments $p = 0.015$. In contrast to baby 1, 6'SL did not induce growth of Actinobacteria in baby 2, to any larger extent.

For fermentations with GM of baby 1 (Table 1) significant differences in the relative abundance of an unassigned Veillonellaceae ($p = 0.0001$) and the OTU *Veillonella* ($p = 0.0001$), were identified between treatments. Both were low abundant OTUs ($<1\%$) in all fermentations, but growing in 3'FL fermentations with 2.8% and 5.5% relative abundance respectively.
**Table 1 Species compositions of HMO fermentations with GM of baby 1.** Microbiota composition of all fermentations (3'SL, 3'FL, 6'SL, FOS and control) for baby 1, as determined by 16S rRNA gene amplicon sequencing.

| BABY1 | | | | Control | FOS | 6SL | 3SL | 3FL | *p*-value |
|---|---|---|---|---|---|---|---|---|---|
| Actinobacteria | Bifidobacteriaceae | *Bifidobacterium* | *bifidum (*99.4)* | 0.03 | 0.05 | 0.21 | 0.21 | 0.05 | 0.28 |
| | Bifidobacteriaceae | *Bifidobacterium* | *longum* | 0.6 | 0.37 | 0.24 | 1.38 | 0.38 | 0.5 |
| | Bifidobacteriaceae | *Bifidobacterium* | *breve (*97.8)* | 2.53 | 3.68 | 15.76 | 15.68 | 1.99 | 0.35 |
| Bacteroidetes | Porphyromonadaceae | *Parabacteroides* | *distasonis* | 1.76 | 1.44 | 24.07 | 14.23 | 31.39 | 0.07 |
| | Porphyromonadaceae | *Parabacteroides* | *gordonii (*96.9)* | 0.01 | 0 | 0.12 | 0.06 | 0.17 | 0.06 |
| | Other | *Other* | *other* | 0.02 | 0.03 | 0.19 | 0.09 | 0.19 | 0.05 |
| | Bacteroidaceae | *Bacteroides* | *ovatus* | 1.31 | 0.32 | 0.84 | 0.78 | 1.08 | 0.65 |
| | Bacteroidaceae | *Bacteroides* | *caccae* | 0.46 | 3.61 | 0.39 | 0.37 | 0.58 | 0.19 |
| | Bacteroidaceae | *Bacteroides* | | 2.14 | 4.68 | 6.67 | 4.62 | 8.63 | 0.64 |
| | Bacteroidaceae | *Bacteroides* | *uniformis* | 0.19 | 0.19 | 0.59 | 0.39 | 0.25 | 0.64 |
| | Bacteroidaceae | *Bacteroides* | *acidifaciens* | 0.37 | 0.14 | 0.21 | 0.22 | 0.26 | 0.76 |
| | Bacteroidaceae | *Bacteroides* | *other* | 0.11 | 0.04 | 0.23 | 0.14 | 0.28 | 0.25 |
| Firmicutes | Lactobacillaceae | *Lactobacillus* | *zeae* | 0 | 0.2 | 0 | 0.11 | 0.01 | 0.75 |
| | Clostridiaceae | *Clostridium* | *other* | 11.15 | 9.6 | 9.81 | 13.51 | 5.54 | 0.85 |
| | Clostridiaceae | *Clostridium* | *perfringens* | 7.35 | 6.38 | 6.54 | 9.12 | 3.67 | 0.85 |
| | Veillonellaceae | | | 0.02 | 0.01 | 0.03 | 0.01 | 2.83 | <0.001 |
| | Veillonellaceae | *Veillonella* | | 0.04 | 0.02 | 0.07 | 0.01 | 5.55 | <0.001 |
| | Clostridiaceae | *Clostridium* | | 3.08 | 2.65 | 5.31 | 9.4 | 0.99 | 0.63 |
| | Peptostreptococcaceae | *Clostridium* | *other* | 0.05 | 0 | 0 | 0.43 | 0 | 0.67 |
| | Other | *Other* | *other* | 0.03 | 0.01 | 0.02 | 0.14 | 0.01 | 0.65 |
| | Enterococcaceae | *Enterococcus* | | 0.53 | 0.09 | 0.16 | 0.24 | 0.15 | 0.58 |
| | Clostridiaceae | | | 0.84 | 0.77 | 1.26 | 2.19 | 0.28 | 0.64 |
| | Clostridiaceae | *Other* | *other* | 0.33 | 0.27 | 0.31 | 0.62 | 0.15 | 0.34 |
| | Enterococcaceae | *Other* | *other* | 0.07 | 0.03 | 0.05 | 0.09 | 0.02 | 0.5 |
| Proteobacteria | Enterobacteriaceae | *Dickeya* | | 0.07 | 0.34 | 0.12 | 0.01 | 0.01 | 0.03 |
| | Enterobacteriaceae | *Enterobacter* | *other* | 0.32 | 0.58 | 0.16 | 0.03 | 0.07 | 0.15 |
| | Enterobacteriaceae | *Klebsiella* | | 19.74 | 23.69 | 6.18 | 0.6 | 1 | 0.18 |
| | Enterobacteriaceae | *Other* | *other* | 3.97 | 3.35 | 1.05 | 0.85 | 0.57 | 0.34 |
| | Enterobacteriaceae | *Erwinia* | *other* | 0.09 | 0.12 | 0.07 | 0.02 | 0.02 | 0.27 |
| | Enterobacteriaceae | | | 42.52 | 37.17 | 19 | 23.97 | 33.64 | 0.4 |
| | Other | *Other* | *other* | 0.12 | 0.04 | 0.06 | 0.11 | 0.1 | 0.54 |

In baby 1 phyla Bacteroidetes was represented by nine OTUs on the species level, with the species *Parabacteroides distasonis* displaying the highest abundances in all HMO fermentations (14–31%) compared to the control and FOS fermentations (1–2%) (Table 1, included all OTUs with 0.03% relative abundance in at least one of the treatments). *Bifidobacterium breve* was highly abundant in the 3'SL and 6'SL fermentations (15–16%) in baby 1, but only constituting 2–4% of the GM in the 6'FL, FOS and control fermentations (Table 1).

Baby 2 harbored four species of bifidobacteria, *B. bifidum, B. adolescentis, B. longum and B. faecale. B. bifidum* was the most abundant bifidobacteria, reaching the highest relative abundance in the 3'FL and 3'SL fermentations (at 5–13%) while constituting less

**Table 2 Species composition of HMO fermentations with GM of baby 2.** Microbiota composition of all fermentations (3'SL, 3'FL, 6'SL, FOS and control) for baby 2 as determined by 16S rRNA gene amplicon sequencing.

| Baby 2 | | | | Control | FOS | 6SL | 3SL | 3FL | p value |
|---|---|---|---|---|---|---|---|---|---|
| Actinobacteria | Bifidobacteriaceae | *Bifidobacterium* | *bifidum* | 0.09 | 0.26 | 0.08 | 13.25 | 7.58 | <0.001 |
| | Bifidobacteriaceae | *Bifidobacterium* | *adolescentis* | 0.04 | 0.17 | 0.03 | 10.37 | 5.64 | <0.001 |
| | Bifidobacteriaceae | *Bifidobacterium* | *longum* | 0.07 | 0.19 | 0.06 | 1.94 | 2.14 | 0.02 |
| | Bifidobacteriaceae | *Bifidobacterium* | *fecale (*97.24)* | 0 | 0 | 0 | 0.15 | 0.12 | 0.04 |
| Firmicutes | Ruminococcaceae | | | 0.21 | 0 | 0 | 0 | 0 | 0.28 |
| | Lachnospiraceae | *Blautia* | | 0.91 | 0 | 0 | 0 | 0 | 0.26 |
| | Staphylococcaceae | *Staphylococcus* | *aureus* | 3.93 | 0 | 0 | 0 | 0 | 0.24 |
| | Lachnospiraceae | | | 0.44 | 0 | 0 | 0 | 0 | 0.23 |
| | Clostridiaceae | *Clostridium* | | 1.57 | 0 | 7.56 | 0 | 0 | 0.23 |
| | Clostridiaceae | | | 0.4 | 0 | 1.75 | 0 | 0 | 0.22 |
| | Enterococcaceae | *Other* | *other* | 0 | 0 | 0.09 | 0.04 | 0 | 0.21 |
| | Clostridiaceae | *Clostridium* | *other* | 0.18 | 0.02 | 0.77 | 0.02 | 0.01 | 0.2 |
| | Clostridiaceae | *Other* | *other* | 0.05 | 0 | 0.15 | 0 | 0 | 0.2 |
| | Enterococcaceae | *Enterococcus* | | 0.01 | 0.02 | 0.72 | 0.3 | 0.02 | 0.29 |
| | Streptococcaceae | *Streptococcus* | | 0 | 0 | 0.01 | 0 | 0 | 0.42 |
| | Lactobacillaceae | *Lactobacillus* | | 0 | 0.01 | 0.05 | 0.65 | 0.01 | 0.62 |
| | Clostridiaceae | *Clostridium* | *perfringens* | 0 | 0.01 | 0.01 | 0.01 | 0.01 | 0.91 |
| Proteobacteria | Enterobacteriaceae | *Erwinia* | *other* | 0.15 | 0.11 | 0.17 | 0.04 | 0.19 | 0.39 |
| | Other | *Other* | *other* | 0.23 | 0.24 | 0.32 | 0.28 | 0.43 | 0.44 |
| | Enterobacteriaceae | *Klebsiella* | | 11.2 | 7.21 | 19.89 | 1.41 | 11.38 | 0.44 |
| | Enterobacteriaceae | *Proteus* | | 0.01 | 0 | 0 | 0 | 0.01 | 0.21 |
| | Enterobacteriaceae | *Enterobacter* | *cloacae* | 0.01 | 0.02 | 0 | 0.07 | 0.13 | 0.07 |
| | Enterobacteriaceae | *Enterobacter* | *other* | 0.33 | 0.48 | 0.36 | 0.09 | 0.38 | 0.13 |
| | Enterobacteriaceae | | | 37.28 | 77.23 | 63.89 | 69.03 | 66.64 | 0.16 |
| | Enterobacteriaceae | *Dickeya* | | 0.05 | 0 | 0.02 | 0 | 0 | 0.29 |
| | Enterobacteriaceae | *Other* | *other* | 3.86 | 13.88 | 3.98 | 2.29 | 5.19 | 0.03 |
| | Enterobacteriaceae | *Enterobacter* | *ludwigii* | 0.02 | 0.05 | 0.01 | 0.01 | 0.01 | 0.04 |
| | Enterobacteriaceae | *Trabulsiella* | | 0.01 | 0.02 | 0.01 | 0.01 | 0.01 | 0.27 |

than 1% in the 6'SL, FOS and control fermentations (Table 2, included all OTUs with at least 0.03% relative abundance in at least one of the treatments). The bifidobacterial species of baby 2 did not seem to ferment 6'SL. This is in contrast to baby 1, where especially *Bifidobacterium breve* increased in relative abundance, when growing with 6'SL supplemented media. 3'FL induced the least bifidogenic effect in baby 1 with small increase in growth of *Bifidobacterium breve*, but had the second largest bifidogenic effect for baby 2 promoting growth of *Bifidobacterium bifidum* (7.6%) and *B. adolescentis* (5.6%).

## SCFAs results

When looking at the SCFAs excluding unknowns the HMO treatments did not differ significantly from the control ($p = 0.35$), but display trends of increased amounts of propionic acid in 3'FL and highest butyric acid in FOS. When subtracting the SCFAs found in the control from all other treatments Fig. 6, it becomes clear that Isovaleric and

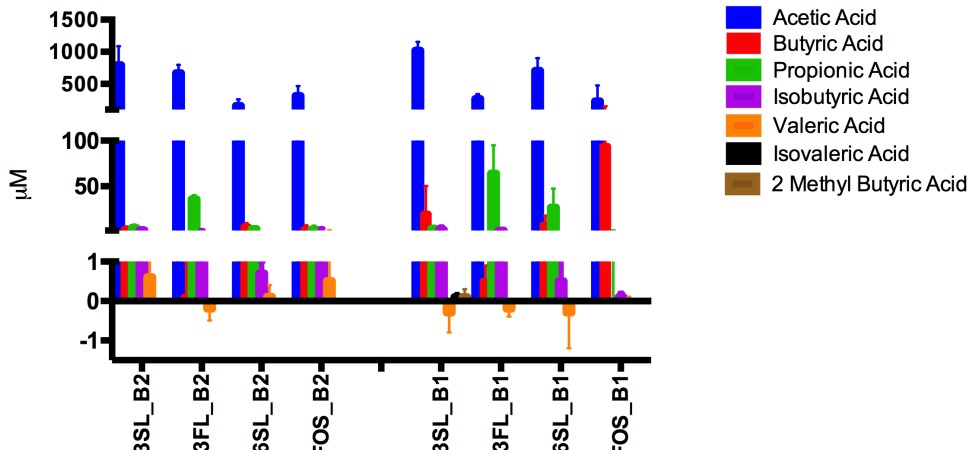

**Figure 6   SCFA content of HMO fermentates.** Quantified SCFAs (μM) produced after 24 h of fermentation of 3'SL, 6'SL, 3'FL and FOS by GM of baby 1 and 2. Amounts displayed represent values after subtraction of SCFAs produced in the control fermentations.

Valeric acid as well as 2-Methyl-Butyric acid mainly originate from substrates derived from media and or feces. For both babies marked substrate effects on SCFAs-production were observed. For baby 1, with significantly higher propionic acid production ($p < 0.006$) and lower amount of butyric acid ($p = 0.01$) in the 3'FL fermentations compared to 3'SL, 6'SL and FOS (Fig. 6). For baby 2, with the same pattern of significantly higher propionic acid production in the 3'FL treatment ($p < 0.0001$) and lower butyric acid production compared to the other treatments. A significant difference was detected for FOS fermentations, with higher butyric acid production in fermentations with baby 1 GM ($p = 0.01$), Fig. 6.

## DISCUSSION

Here we describe a novel, low-volume *in vitro* colon model with increased throughput.

CoMiniGut's automated pH control set-up allows a flexible experimental design, with optional settings for e.g., the simulation of only one colon region within the vessel or—as presented in this study—the passage through the colon simulated by a pH increment over time. Sampling of fermentate is facilitated at all times through a needle inlet. Similarly, standard media can be altered and different feeds can be added gradually or at specific time points in a programmed manner via a syringe pump.

Due to the small amount of fecal sample needed for the inoculations of each experiment in CoMiniGut (250 μg of the fecal matter for the inoculation of five reactors), fecal cryo-stock libraries can be constructed, that way facilitating multiple experimental repetitions, something that is not possible, when fresh inoculum is used. Frozen stocks furthermore facilitate an increased comparability between the donors, as samples can be processed consistently from delivery to inoculation. A synchronized delivery of fresh fecal material especially from e.g., infants is difficult to achieve. No significant differences in culturable cell counts were observed between fresh fecal samples and the frozen samples, using the same conditions as for CoMiniGut fecal inoculum implemented in this study (Fig. S3)

Due to its small working volume CoMiniGut will also allow for the investigation of expensive compounds or novel synthesized materials only available in small amounts, which otherwise would be very costly or simply impossible. For the conduction of a fermentation at 1% (w/v) only 50 mg of substrate is needed per fermentation reaction. The requirement of only small amounts of substrate and fecal material facilitates the screening of a plethora of different bioactives, in different doses and combinations, for one or multiple fecal donors.

Using inulin and lactulose as substrate for fermentations inoculated with fecal slurry from two adult donors significant substrate as well as inter-individual differences in microbiota composition and SCFAs-profiles were demonstrated. *Rycroft et al. (2001)* investigated *in vitro* simulated colon fermentations of several prebiotics including inulin and lactulose using fluorescence *in situ* hybridization (FISH) to investigate the effect on microbiota composition. Similar to *Rycroft et al. (2001)* we observed higher acetic acid levels after 24 h of fermentation of lactulose *in vitro* compared to inulin fermentations, though in the present study the differences in acetic acid production were not significant. Venema et al. also investigated the effects of lactulose on the composition of GM and SCFAs in human volunteers and using the TIM-2 *in vitro* model. They, similar to the present study, found low levels of butyrate production when fermenting lactulose *in vitro* (*Venema et al., 2003*). Further, when comparing the ratios of the SCFAs (butyrate, propionate and acetate) it is seen that the ratios of SCFAs found in our study are comparable to those observed by *Venema et al. (2003)* in the adapted microbial communities. This shows that SCFAs-results obtained using the CoMiniGut are comparable to existing *in vitro* models with a higher working volumes and lower throughput. The model further facilitated the fermentations of the rare and expensive HMOs only available in small amounts with two infant GMs. We have extracted inter-individual difference of HMO utilization capacity especially with regards to 6-sialyated HMOs.

The site of sialylation 3 vs 6′SL impacts the degree of gut microbial growth and its utilization depends on the metabolic capacity and the ability to utilize it within a given microbial consortium. In this study we have found the strongest average bifidogenic effect to be induced by 3′SL overall when looking at both infants. The lack of a bifidogenic effect in the 6′SL-supplemented fermentations of baby 2 compared to the significant population ($\approx$16%) of Actinobacteria in 6′SL fermentations of GM from baby 1 (Fig. 5) is striking.

The GM of both infants harbored different bifidobacteria species. While both babies harboured members of *Bifidobacterium bifidum* and *B. longum*, the GM of baby 1 also harboured *Bifidobacterium breve*, whereas baby 2 harboured *B. adolescentis* and *Bifidobacterium faecale*. None of the bifidobacterial species present in the baby 1 GM did increase in relative abundance during 6′SL fermentations (Table 1), whereas *Bifidobacterium breve* (baby 2) grew in 6′SL fermentations. Different Bifidobacterial species have varying physiologies leading to varying capacities of HMO utilization, and even strain dependent utilization of HMOs has for instance been described for *Bifidobacterium breve* (*LoCascio et al., 2007*; *LoCascio et al., 2009*; *Ruiz-Moyano et al., 2013*). It has been suggested that *B. longum* subsp. *infantis* has an inherent and constitutive ability to process sialyated compounds (*Yu, Chen & Newburg, 2013*). Nevertheless the *B. longum* strain of baby 2 did

not seem to utilize 6′SL, which might be due to the specific site of sialylation. Another explanation for the observed discrepancies is a syntrophic interaction between members of the phyla Bacteroidetes, especially *Parabacteroides distasonis,* which are present in baby 1 but barely present in baby 2, and bifidobacteria.

Also, the utilization of 3′FL varies between baby 1 and baby 2. It is possible that this is due to different strains of the species *B. bifidum* and *B. longum* in baby 2 vs. baby 1. Nevertheless, it is also possible that the discrepancies between baby 1 and baby 2 relate to the difference in overall microbial community and the presence of Bacteroidetes, which are capable of HMO utilization. *Yu, Chen & Newburg (2013)* reported that when supplemented with 3′FL, all *Bifidobacterium* spp. and *Bacteroides* spp. displayed appreciable induction of Alfa-L-Fucosidase, AFU activity and consumed 40% or greater of the 3′FL in their study. In the present study it is hence likely that Bacteroidetes thrive on 3′FL and compete with the bifidobacterial strains explaining comparably lower bifidobacterial abundance in baby 1, when growing together with 3′FL utilizing Bacteroidetes.

## CONCLUSIONS

The construction of fecal libraries of specific target groups such as babies, healthy adults, elderly, diabetics, or obese donors allows for the assessment of single or multiple dietary components, and their impact on individual gut microbial populations and subsequent changes to their metabolism. In combination with advanced high throughput sequencing protocols CoMiniGut constitutes an excellent tool for the investigation of not only bacterial species but also interactions between bacteriophages and eukaryotic microorganisms and various microbe-microbe and microbe-bioactive interactions *in vitro*. In this study we show superior advantages of the newly developed *in vitro* colon model with regards to its throughput, reproducibility and its potential application for the investigation of rare and expensive compounds. To conclude, the CoMiniGut facilitates next generation *in vitro* colon simulations with high statistical inference and simultaneous reduction of resources.

## ACKNOWLEDGEMENTS

Rasmus Riemer Jakobsen is acknowledged for excellent technical assistance with the evaluation of cryopreservation protocols.

### Funding

This project was funded by the Danish Council for Independent Research (DFF), Technology and Production, under the grant DFF-1335-00177. The funders had no role in study design, data collection and analysis, decision to publish, or preparation of the manuscript.

### Grant Disclosures

The following grant information was disclosed by the authors:

Danish Council for Independent Research (DFF), Technology and Production: DFF-1335-00177.

## Competing Interests

The authors declare there are no competing interests.

## Author Contributions

- Maria Wiese conceived and designed the experiments, performed the experiments, analyzed the data, contributed reagents/materials/analysis tools, wrote the paper, prepared figures and/or tables, reviewed drafts of the paper.
- Bekzod Khakimov analyzed the data, reviewed drafts of the paper.
- Sebastian Nielsen contributed reagents/materials/analysis tools.
- Helena Sørensen performed the experiments.
- Frans van den Berg and Dennis Sandris Nielsen conceived and designed the experiments, contributed reagents/materials/analysis tools, wrote the paper, reviewed drafts of the paper.

## Human Ethics

The following information was supplied relating to ethical approvals (i.e., approving body and any reference numbers):

Sampling and use of fecal material for inoculation of the CoMiniGut fermentations has been approved by the Ethical Committee (E) for the Capital Region of Denmark (H-15001754).

## Data Availability

The raw data has been provided in the Supplemental Files.

## Supplemental Information

Supplemental information for this article can be found online at http://dx.doi.org/10.7717/peerj.4268#supplemental-information.

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
