# Peer review of "CoMiniGut—a small volume in vitro colon model for the screening of gut microbial fermentation processes"

_PeerJ, doi:10.7717/peerj.4268_

## Round 0.1 · original submission · Major Revisions

· Academic Editor

Major Revisions

While I agree with both reviewers, that the system described in your manuscript is a very interesting addition to the currently available suite of in vitro fermentation models, there are a number of issues that should be addressed in a revised manuscript. As pointed out by both reviewers, some of the technical aspects of the model could be explored more, including sampling at different time points as well as fed-batch operation.
Also, some details of used methods are lacking as indicated in more detail in the reviewers' reports.

·

Basic reporting

The paper is well writen and the English is, as far as I can evaluate, appropriate.
Appropriate references have been used and the structure is clear.
However, the article addresses basically two topics; the CoMiniGut and the in vitro fermentation of selected substrates. It would better to discuss these topics in more detail (in particular the CoMiniGut) separately

Experimental design

The manuscript describes the targets and these are in scope with the Journal. The research question is well defined, but not fully achieved as far as the CoMiniGut is concerned. Methods are described in sufficient detail.

Validity of the findings

The novelty is not as much as it could be. In the end, the simulations are pH-controlled faecal batch fermentations, which are assessed after 24 h. While this generates interesting preliminary data on what groups in the faecal microbiota are or are not able to grow in the presence of the tested HMOs, this could have been investigated in more depth as the authors actually suggest themselves in the discussion of the model.

Reviewer 2 ·

Basic reporting

The manuscript of Weise et al develops a low volume batch system for modelling microbial changes in the gut. The systems low volumes lends itself to sparse compounds. This could therefore prove a useful model. I have a few associated questions for the authors.
Line 115 - FOS often isolated from vegetable sources, rather than enzyme can extraction.

Experimental design

I note that fecal stocks frozen in glycerol; has this been tested to see the loss in viability brought about by freezing? Were fresh samples run side by side?
Were timings kept the same for infant and adult models? Literature suggests adult RT is longer than 24hrs.
Were samples taken at the different times to echo the differences caused by changing the pH? Did this match what would be expected at the colonic regions?
Which time-points chosen - could you have missed peaks in microbial growth?

Validity of the findings

The points raised in 2 would impact on the validity of the findings. However, the authors have related the findings to literature

---

## Round 0.2 · Minor Revisions

· Academic Editor

Minor Revisions

Please see a few minor notes of reviewer 1 regarding one of the references, and UK vs US English.

·

Basic reporting

The authors have answered the concerns I raised in a satisfactory way.
However, I noted that in the description of the basal colon medium, no reference is made to Saulnier et al. 2008 eventhough the article is included in the references, it is not included in the text; I guess it should be mentioned in lines 186-187. Furthermore, I noted there are actually some words in UK English while the overall paper seems to be meant to be written in US English; e.g. 'harboured' (lines 432-434)

Experimental design

No further comments.

Validity of the findings

No further comments.

Comments for the author

No further comments.

---

## Round 0.3 · accepted · Accept

· Academic Editor

Accept

Thank you for efficiently turning around reviewers' comments during two rounds of revisions. Very interesting paper describing a new in vitro gut model that is an important contribution to the field.